# ANALYZING DIFFUSION AS SERIAL REPRODUCTION

## ABSTRACT

Diffusion models are a class of generative models that learn to synthesize samples by inverting a diffusion process that gradually maps data into noise. While these models have enjoyed great success recently, a full theoretical understanding of their observed properties is still lacking, in particular, their weak sensitivity to the choice of noise family and the role of adequate scheduling of noise levels for good synthesis. By identifying a correspondence between diffusion models and a well-known paradigm in cognitive science known as serial reproduction, whereby human agents iteratively observe and reproduce stimuli from memory, we show how the aforementioned properties of diffusion models can be explained as a natural consequence of this correspondence. We then complement our theoretical analysis with simulations that exhibit these key features. Our work highlights how classic paradigms in cognitive science can shed light on state-of-the-art machine learning problems.

## 1 INTRODUCTION

Diffusion models are a class of deep generative models that have enjoyed great success recently in the context of image generation (Sohl-Dickstein et al., 2015; Ho et al., 2020; Song & Ermon, 2019; Rombach et al., 2022; Ramesh et al., 2022), with some particularly impressive text-to-image applications such as DALL-E 2 (Ramesh et al., 2022) and Stable Diffusion (Rombach et al., 2022). The idea behind diffusion models is to learn a data distribution by training a model to invert a diffusion process that gradually destroys data by adding noise (Sohl-Dickstein et al., 2015). Given the trained model, sampling is then done using a sequential procedure whereby an input signal (e.g., a noisy image) is iteratively denoised at different noise levels which, in turn, are successively made finer until a sharp sample is generated. Initially, the noise family was restricted to the Gaussian class (Sohl-Dickstein et al., 2015; Song & Ermon, 2019; Ho et al., 2020) and the process was understood as a form of Langevin dynamics (Song & Ermon, 2019). However, recent work showed that this assumption can be relaxed substantially (Bansal et al., 2022; Daras et al., 2022) by training diffusion models with a wide array of degradation families. One feature of this work is that it highlights the idea that sampling (i.e. synthesis) can be thought of more generally as an alternating process between degradation and restoration operators (Bansal et al., 2022). This in turn calls into question the theoretical understanding of these models and necessitates new approaches.

A hint at a strategy for understanding diffusion models comes from noting that the structure of the sampling procedure in these generalized models (i.e., as a cascade of noising-denoising units), as well as its robustness to the choice of noise model, bears striking resemblance to a classic paradigm in cognitive science known as *serial reproduction* (Bartlett & Bartlett, 1995; Xu & Griffiths, 2010; Jacoby & McDermott, 2017; Langlois et al., 2021). In a serial reproduction task, participants observe a certain stimulus, e.g., a drawing or a piece of text, and then are asked to reproduce it from memory (Figure 1A). The reproduction then gets passed on to a new participant who in turn repeats the process and so on. The idea is that as people repeatedly observe (i.e., encode) a stimulus and then reproduce (i.e., decode) it from memory, their internal biases build up so that the asymptotic dynamics of this process end up revealing their inductive biases (or prior beliefs) with regard to that stimulus domain. By modeling this process using Bayesian agents, Xu & Griffiths (2010) showed that the process can be interpreted as a Gibbs sampler, and more so, that the stationary behavior of this process is in fact independent of the nature of cognitive noise involved, making serial reproduction a particularly attractive tool for studying human priors (Figure 1B). The main contribution of the present paper is to make the correspondence between diffusion and serial reproduction precise

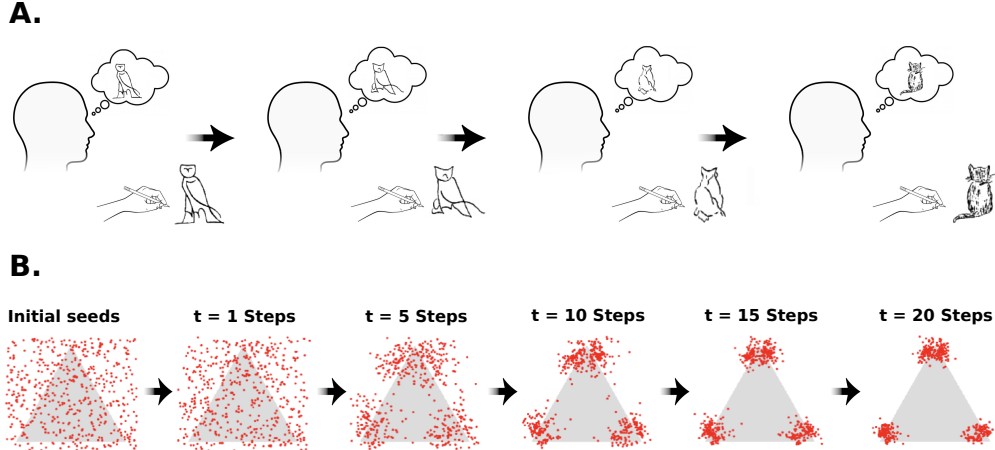

Figure 1: Serial reproduction paradigm. **A.** Participants observe (encode) a stimulus and then try to reproduce (decode) it from memory. As the process unfolds, the generated samples gradually change until they become consistent with people's priors. Drawings are reproduced from Bartlett & Bartlett (1995). **B.** Data from a real serial reproduction task reproduced from Langlois et al. (2021). Participants observed a red dot placed on a background image (here a triangle) and were instructed to reproduce the location of that dot from memory. The new dot location then gets passed to a new participant who in turn repeats the task. The initial uniform distribution gets transformed into a highly concentrated distribution around the triangle's corners, capturing people's visual priors.

and show how the observed properties of diffusion models, namely, their robustness to the choice of noise family, and the role of adequate noise level scheduling for good sampling, jointly arise as a natural consequence. The paper proceeds as follows. In Section 2 we review the mathematical formulation of serial reproduction and diffusion models and set the ground for the analysis that follows. In Section 3, we establish a correspondence between sampling in diffusion models and serial reproduction and show how it explains key properties of these models. In Section 4 we complement our theoretical analysis with simulations, and then conclude with a discussion in Section 5.

## 2 BACKGROUND

### 2.1 SERIAL REPRODUCTION

We begin with a brief exposition of the mathematical formulation of the serial reproduction paradigm (Xu & Griffiths, 2010; Jacoby & McDermott, 2017). A *serial reproduction process* is a Markov chain over a sequence of stimuli (images, sounds, text, etc.) $x_0 \rightarrow x_1 \rightarrow \cdots \rightarrow x_t \rightarrow \ldots$ where the dynamics are specified by the encoding-decoding cascade of a Bayesian agent with some prior $\pi(x)$ and a likelihood model $p(\hat{x}|x)$. The prior captures the previous experiences of the agent with the domain (i.e., their inductive bias), and the likelihood specifies how input stimuli $x$ map into noisy percepts $\hat{x}$ (e.g., due to perceptual, production or cognitive noise). Specifically, given an input stimulus $x_t$, the agent *encodes* $x_t$ as a noisy percept $\hat{x}_t$, and then at reproduction *decodes* it into a new stimulus $x_{t+1}$ by sampling from the Bayesian posterior (a phenomenon known as probability matching) (Griffiths & Kalish, 2007),

$$p(x_{t+1}|\hat{x}_t) = \frac{p(\hat{x}_t|x_{t+1})\pi(x_{t+1})}{\int p(\hat{x}_t|\tilde{x}_{t+1})\pi(\tilde{x}_{t+1})d\tilde{x}_{t+1}}. \tag{1}$$

The generated stimulus $x_{t+1}$ is then passed on to a new Bayesian agent with similar prior and likelihood who in turn repeats the process and so on. From here, we see that the transition kernel of

the process can be derived by integrating over all the intermediate noise values against the posterior

$$
\begin{aligned}
p(x_{t+1}|x_t) &= \int p(x_{t+1}|\hat{x}_t)p(\hat{x}_t|x_t)d\hat{x}_t \\
&= \int \frac{p(\hat{x}_t|x_{t+1})p(\hat{x}_t|x_t)}{\int p(\hat{x}_t|\tilde{x}_{t+1})\pi(\tilde{x}_{t+1})d\tilde{x}_{t+1}}\pi(x_{t+1})d\hat{x}_t.
\end{aligned}
\tag{2}
$$

Crucially, by noting that $\tilde{x}_{t+1}$ is a dummy integration variable we see that the prior $\pi(x)$ satisfies the *detailed-balance condition* with respect to this kernel

$$
p(x_{t+1}|x_t)\pi(x_t) = p(x_t|x_{t+1})\pi(x_{t+1}).
\tag{3}
$$

This in turn implies that the prior $\pi(x)$ is the stationary distribution of the serial reproduction process irrespective of the noise model $p(\hat{x}|x)$, so long as it allows for finite transition probabilities between any pair of stimuli to preserve ergodicity (Xu & Griffiths, 2010; Griffiths & Kalish, 2007). This insensitivity to noise is what makes serial reproduction a particularly attractive tool for studying inductive biases in humans (Figure 1B). It is also worth noting that another way to derive these results is by observing that the full process over stimulus-percept pairs $(x, \hat{x})$ whereby one alternates between samples from the likelihood $p(\hat{x}|x)$ and samples from the posterior $p(x|\hat{x})$ implements a Gibbs sampler from the joint distribution $p(x, \hat{x}) = p(\hat{x}|x)\pi(x)$.

## 2.2 DIFFUSION MODELS

We next review the basics of diffusion models and set the ground for the analysis in the next section. Following (Sohl-Dickstein et al., 2015; Ho et al., 2020), a diffusion model is a generative model that learns to sample data out of noise by inverting some specified *forward process* $q(x_0, \ldots, x_T)$ that gradually maps data $x_0 \sim q_d(x_0)$ to noise $x_T \sim q_n(x_T)$, where $q_d(x)$ and $q_n(x_T)$ are given data and noise distributions, respectively. Such forward process can be implemented as a Markov chain

$$
q(x_0, \ldots, x_T) = q_d(x_0)\prod_{t=1}^{T} q(x_t|x_{t-1})
\tag{4}
$$

with some pre-specified transition probabilities $q(x_t|x_{t-1}) = T_{q_n}(x_t|x_{t-1}; \beta_t)$ where $T_{q_n}$ is a noise (diffusion) kernel and $\beta_t$ being some diffusion parameter for which the noise distribution $q_n$ is stationary, i.e., $\int T_{q_n}(y|x)q_n(x)dx = q_n(y)$. This ensures that for a sufficiently large time $t = T$ we are guaranteed to transform $q_d(x)$ into $q_n(x)$. A common explicit example of this is a Gaussian kernel $T_{q_n}(x_t|x_{t-1}; \beta_t) = \mathcal{N}(x_t; \sqrt{1-\beta_t}x_{t-1}, \beta_t I)$ where $I$ is the identity matrix, however we will not assume that. The inversion is then done by solving a variational problem with respect to a trainable *reverse process* $p_\theta(x_0, \ldots, x_T)$ which itself is assumed to be Markov

$$
p_\theta(x_0, \ldots, x_T) = p(x_T)\prod_{t=1}^{T} p_\theta(x_{t-1}|x_t)
\tag{5}
$$

where $p(x_T) = q_n(x_T)$, that is, the reverse process starts from noise and iteratively builds its way back to data. Since we are interested in the optimal structure of $p_\theta$ we will suppress $\theta$ in what follows. The reverse process induces a probability distribution over data $p(x_0)$ by marginalizing Equation 5 over all $x_{t>0}$. The variational objective is then given by a bound $K$ on the log-likelihood of the data under the reverse process $L = -\int q_d(x_0)\log p(x_0)dx_0$ and can be written as (Sohl-Dickstein et al., 2015)

$$
L \geq K = \sum_{t=1}^{T}\int q(x_0, \ldots, x_T)\log\left[\frac{p(x_{t-1}|x_t)}{q(x_t|x_{t-1})}\right]dx_0\ldots dx_T - H_{q_n}
\tag{6}
$$

where $H_{q_n}$ is the entropy of the noise distribution $q_n$ which is a constant. Finally, by defining the forward posterior

$$
q(x_{t-1}|x_t) \equiv \frac{q(x_t|x_{t-1})q(x_{t-1})}{\int q(x_t|\tilde{x}_{t-1})q(\tilde{x}_{t-1})d\tilde{x}_{t-1}} = \frac{q(x_t|x_{t-1})q(x_{t-1})}{q(x_t)}
\tag{7}
$$

where $q(x_{t-1})$ and $q(x_t)$ are the marginals of the forward process (Equation 4) at steps $t-1$ and $t$, respectively, we can rewrite $K$ as (see Appendix A)

$$
K = -\sum_{t=1}^{T}\mathbb{E}_{x_t \sim q}D_{KL}\left[q(x_{t-1}|x_t)||p(x_{t-1}|x_t)\right] + C_q
\tag{8}
$$

where $D_{KL}$ is the Kullback-Leibler divergence and $C_q$ is a constant. While Equation 8 is not necessarily the most tractable form of the bound $K$, it will prove useful in the next section when we make the connection to serial reproduction (see Appendix A, Eqs. 17-26 in Ho et al. (2020) for a similar derivation).

Before proceeding to the next section, it is worth pausing for a moment to review the existing theoretical interpretations of diffusion models and the challenges they face. Two general formulations of diffusion models exist in the literature, namely, Denoising Diffusion Probabilistic Models (DDPMs) (Ho et al., 2020; Sohl-Dickstein et al., 2015) which adopt a formulation similar to the one used here, and Score-Based Models (Song & Ermon, 2019; Daras et al., 2022) which learn to approximate the gradient of the log-likelihood of the data distribution under different degradations, also known as the score of a distribution, and then incorporate that gradient in a stochastic process that samples from the data distribution. These formulations are not entirely independent and in fact have been shown to arise from a certain class of stochastic differential equations (Song et al., 2020). Importantly, these analyses often assume that the structure of noise is Gaussian, either to allow for tractable expressions for variational loss functions (Sohl-Dickstein et al., 2015), or as a way to link sampling to well-known processes such as Langevin dynamics (Song & Ermon, 2019). This theoretical reliance on Gaussian noise has been recently called into question as successful applications of diffusion models with a wide variety of noise classes were demonstrated empirically (Bansal et al., 2022; Daras et al., 2022). We seek to remedy this issue in the next section.

## 3 DIFFUSION SAMPLING AS SERIAL REPRODUCTION

### 3.1 MATHEMATICAL DERIVATION

As noted in the introduction, the sampling procedure for diffusion models is strikingly similar to the process of serial reproduction. In what follows we will make this statement more precise and show how it allows to explain key features of diffusion models. First, observe that the non-negativity of the KL divergence implies that the bound in Equation 8 is maximized by the solution

$$p(x_{t-1}|x_t) = q(x_{t-1}|x_t) = \frac{q(x_t|x_{t-1})q(x_{t-1})}{\int q(x_t|\tilde{x}_{t-1})q(\tilde{x}_{t-1})d\tilde{x}_{t-1}}. \tag{9}$$

In other words, what the diffusion model is approximating at each step $t-1$ is simply a Bayesian posterior with the diffusion kernel $q(x_t|x_{t-1}) = T_{q_n}(x_t|x_{t-1}; \beta_t)$ serving as likelihood and the forward marginal at step $t-1$, namely $q(x_{t-1})$, serving as a prior. For clarity, in what follows we will denote the optimal posterior distribution at step $t-1$ (Equation 9) as $p_{t-1}(x|y)$ and the marginal at step $t-1$ as $q_{t-1}(x)$.

Next, to make contact with the recent literature that extends diffusion models to generalized noise families which decompose sampling into a process that alternates between restoration and degradation (Bansal et al., 2022; Daras et al., 2022) we define the *sampling process* for a diffusion model as a Markov sequence $x_T \to \hat{x}_T \to x_{T-1} \to \cdots \to x_t \to \hat{x}_t \to x_{t-1} \to \cdots \to x_0$, where $x_T \sim q_n(x_T)$, $\hat{x}_t$ is a noisy version of $x_t$ under the noise kernel $T_{q_n}(\hat{x}_t|x_t; \beta_t)$, and the transition $\hat{x}_t \to x_{t-1}$ is done by denoising using the posterior $p_{t-1}$. While this might seem at first to be at odds with the standard definition of sampling in Gaussian DDPMs whereby sampling is done by iterative applications of the posterior in Equation 9, we will show in Section 3.2 that our definition is in fact equivalent. From here, the transition kernel for the sampling process at step $t-1$ which we denote by $p_{s,t-1}(x_{t-1}|x_t)$ is given by

$$\begin{aligned} p_{s,t-1}(x_{t-1}|x_t) &= \int p_{t-1}(x_{t-1}|\hat{x}_t)T_{q_n}(\hat{x}_t|x_t; \beta_t)d\hat{x}_t \\ &= \int \frac{T_{q_n}(\hat{x}_t|x_{t-1}; \beta_t)T_{q_n}(\hat{x}_t|x_t; \beta_t)}{\int T_{q_n}(\hat{x}_t|\tilde{x}_{t-1}; \beta_t)q_{t-1}(\tilde{x}_{t-1})d\tilde{x}_{t-1}}q_{t-1}(x_{t-1})d\hat{x}_t. \end{aligned} \tag{10}$$

This kernel has an identical structure to the serial reproduction kernel in Equation 2, and as such, the forward marginal $q_{t-1}$ satisfies detailed balance with respect to it. In other words, we have a set of detailed-balance conditions given by

$$p_{s,t-1}(x_t|x_{t-1})q_{t-1}(x_{t-1}) = p_{s,t-1}(x_{t-1}|x_t)q_{t-1}(x_t). \tag{11}$$

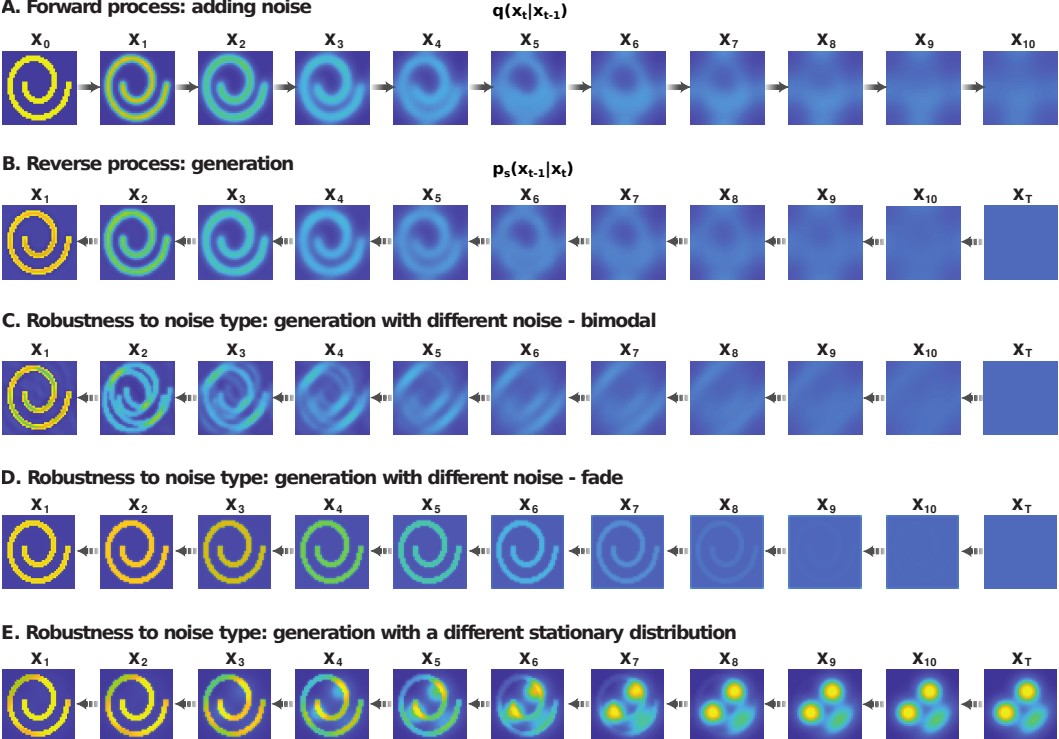

Figure 2: Simulation results for the optimal sampling process. Data was generated by sampling two dimensional vectors from a Swiss-roll distribution. We considered different diffusion noise families and computed their corresponding reverse sampling processes. Each image corresponds to the distribution of samples at a given iteration in a process. **A.** Forward process for a Gaussian noise kernel. **B.** The corresponding reverse sampling process. **C.** Reverse sampling process for a bimodal kernel. **D.** Reverse sampling for a fade-type noise with uniform stationary distribution. **E.** Reverse sampling for a fade-type noise with mixture stationary distribution.

In particular, the last of these $p_{s,0}$ satisfies this condition for the true data distribution since by definition $q_0(x) = q_d(x)$. Now, observe that unlike the case in serial reproduction where we had a single distribution $\pi(x)$ satisfying detailed balance at all steps, here we have a sequence of such distributions. Nevertheless, we can still analyze the induced sampling distribution $p_s(x_0)$ in light of Equation 11. Indeed, by substituting the detailed-balance conditions we have

$$
\begin{aligned}
p_s(x_0) &= \int p_{s,0}(x_0|x_1)\dots p_{s,T-1}(x_{T-1}|x_T)q_n(x_T)dx_{1\dots T} \\
&= q_d(x_0)\int p_{s,0}(x_1|x_0)\frac{q_1(x_1)}{q_d(x_1)}\dots p_{s,T-1}(x_T|x_{T-1})\frac{q_n(x_T)}{q_{T-1}(x_T)}dx_{1\dots T}.
\end{aligned}
\tag{12}
$$

From here we see that the performance of $p_s(x_0)$ as a good approximator of the true data distribution $q_d(x_0)$ critically depends on whether the integral on the right-hand-side of Equation 12 sums up to one. Observe that the integral can be evaluated step-by-step by first integrating over $x_T$, then $x_{T-1}$ and down to $x_1$. For the $x_T$ integral we have

$$
\int p_{s,T-1}(x_T|x_{T-1})\frac{q_n(x_T)}{q_{T-1}(x_T)}dx_T = \int \frac{T(\hat{x}|x_{T-1};\beta_T)}{q_T(\hat{x})}\int T(\hat{x}|x_T;\beta_T)q_n(x_T)dx_Td\hat{x}. \tag{13}
$$

Now, using the fact that $q_n$ is stationary with respect to $T_{q_n}$ and that by construction $q_T(x) = q_n(x)$, the rightmost integral and the denominator cancel out and the remaining integral over $\hat{x}$ integrates to 1. Going one step further to the integral over $x_{T-1}$ (and its own latent $\hat{x}$) we have

$$
\int p_{s,T-2}(x_{T-1}|x_{T-2})\frac{q_{T-1}(x_{T-1})}{q_{T-2}(x_{T-1})}dx_{T-1} = \int \frac{T(\hat{x}|x_{T-2};\beta_{T-1})}{q_{T-1}(\hat{x})}\int T(\hat{x}|x_{T-1};\beta_{T-1})q_{T-1}(x_{T-1}).
\tag{14}
$$

Unlike before, we are no longer guaranteed stationarity for $q_{T-1}$. One trivial solution for this is to use a very strong (abrupt) diffusion schedule $\{\beta_t\}$ such that the marginals behave like $q_0 = q_d$ and $q_{t>0} = q_n$, that is, within one step we are pretty much at noise level. From the perspective of the Bayesian posterior in Equation 9, this limit corresponds to the case where the Bayesian inversion simply ignores the very noisy input and instead relies on its learned prior which happens to be the data distribution for $p_0$. Such a solution, however, is not really feasible in practice because it means that the denoising network that approximates the final Bayesian posterior $p_0$ must learn to map pure noise to true data samples in one step, but this is precisely the problem that we're trying to solve. We therefore exclude this solution.

Luckily, there is another way around this problem, namely, if the diffusion parameter $\beta_{T-1}$ is chosen such that it changes $q_{T-1}$ only by a little so that it is *approximately stationary*, that is,

$$q_{T-1}(\hat{x}) \approx \int dx_{T-1} T(\hat{x}|x_{T-1}; \beta_{T-1}) q_{T-1}(x_{T-1}), \qquad (15)$$

then the integral in Equation 14 will again be close to 1, as the denominator cancels out with the rightmost integral. By induction, we can repeat this process for all lower levels and conclude that

$$p_s(x_0) \approx q_d(x_0) \qquad (16)$$

Crucially, under this schedule denoising networks only need to learn to invert between successive levels of noise. This suggests that the structure of the optimal sampling processes allows one to trade a hard abrupt noise schedule with a feasible smooth one by spreading the reconstruction complexity along the diffusion path in a divide-and-conquer fashion.

To summarize, we have shown that the sampling process of an optimal diffusion model approximates the true data distribution irrespective of the choice of noise family, so long as the noise schedule $\{\beta_t\}$ is chosen such that it alters the input distribution gradually. This result is consistent with recent work suggesting that a good scheduling heuristic minimizes the sum of Wasserstein distances between pairs of distributions along the diffusion path of the forward process (Daras et al. (2022); see also Dhariwal & Nichol (2021)). It also explains the necessary thinning in $\beta_t$ for later stages of the synthesis where the marginal becomes more and more structured. In the Section 4, we provide empirical support for this theoretical analysis.

## 3.2 CLARIFYING THE CONNECTION TO PREVIOUS WORK

Before moving on to the empirical analysis, we complete our theoretical derivation by showing that our definition of the sampling process is equivalent to the one used in the well-studied Gaussian DDPM formalism of Ho et al. (2020). As noted in that paper (Section 3.2 of Ho et al. (2020)), the posterior is taken to be of the form $p(x_{t-1}|x_t) = \mathcal{N}(x_{t-1}; \mu(x_t, t), \Sigma(x_t, t))$ with $\Sigma(x_t, t) = \sigma_t^2 I$, where $\mu(x_t, t)$ is a trainable function and $I$ is the identity matrix. This can be also written as $x_{t-1} = \mu(x_t, t) + \sigma_t z$ where $z \sim \mathcal{N}(0, I)$. In other words, the posterior is given by a Gaussian distribution around a function of the input $\mu(x_t, t)$ with some diagonal covariance matrix with variance $\sigma_t^2$. Intuitively, equivalence then follows from the fact that introducing an additional noising step in the sampling process is simply adding Gaussian noise to a Gaussian posterior which corresponds to a redefinition of the mean and variance parameters (which are design parameters; Dhariwal & Nichol (2021)). More explicitly, our sampling process is defined as $x_T \to \hat{x}_T \to x_{T-1} \to \cdots \to x_0$, that is, we start from an initial sample $x_T$ and then successively add noise to it and then denoise it with the posterior. Now, since the initial $x_T \sim q_n(x_T)$ is sampled from the stationary noise distribution, adding noise to it (i.e., transitioning to $\hat{x}_T$) does not change its distribution so we can equivalently start by denoising $x_T$ using the posterior (as in DDPM) and then adding noise and so on. Mathematically, this corresponds to applying to the generated posterior sample (i.e., the denoised $x_T$) a generic noisy transformation of the form $x \to \alpha x + \sigma z'$ where $z' \sim \mathcal{N}(0, I)$ where $\alpha$ and $\sigma$ are some scaling and variance parameters (similar to Equation 2 in Ho et al. (2020)). Now, when combined with the Gaussian posterior above this yields $x_{T-1} = \alpha \mu(x_T, T) + \alpha \sigma_T z + \sigma z'$ which corresponds to $p(x_{T-1}|x_T) = \mathcal{N}(x_{T-1}; \alpha \mu(x_T, T), (\alpha^2 \sigma_T^2 + \sigma^2)I)$, but this is equivalent to the formula used in Ho et al. (2020) up to a redefinition of the mean and variance, namely, $\mu_T(x_T, T) \to \alpha \mu(x_T, T)$ and $\sigma_T^2 \to \alpha^2 \sigma_T^2 + \sigma^2$. The same holds for all subsequent steps which, as in the case of DDPM, terminate with a final application of the posterior denoiser. Thus, we see that the two definitions are equivalent up to a redefinition of the trainable posterior parameters.

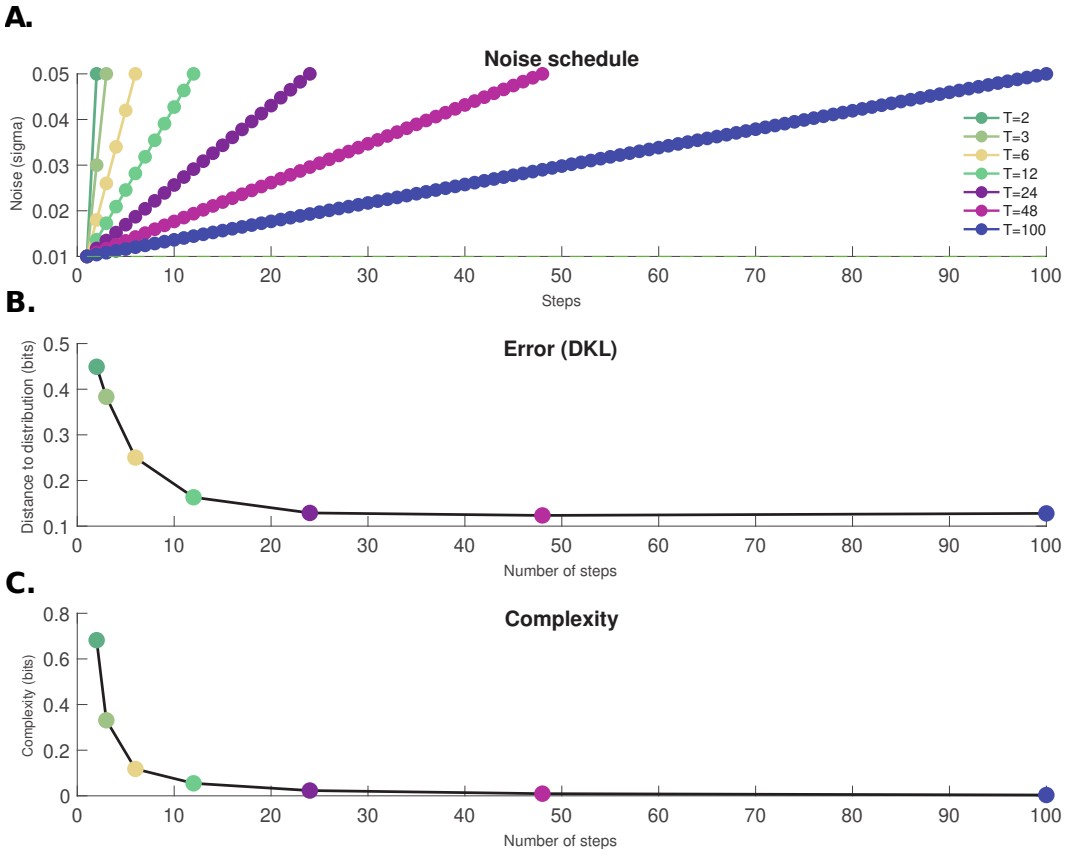

Figure 3: Effect of noise schedule on reconstruction error. **A.** Different noise injection schedules for a Gaussian noise kernel. **B.** Reconstruction error as measured by the KL divergence between the generated distribution and the true data distribution. **C.** A measure of inversion "complexity" as quantified by the maximum KL divergence between two consecutive marginals along the forward diffusion path.

## 4 SIMULATIONS

To test the theoretical findings of the previous section, we selected a simulation setup in which the Bayesian posteriors are tractable and can be computed analytically. Specifically, similar to Sohl-Dickstein et al. (2015), we considered a case where the data is given by two-dimensional vectors $(x, y)$ in the range $-0.5 \leq x, y \leq 0.5$ that are sampled from a Swiss-roll-like data distribution. For tractability, we discretized the space into a finite grid of size $41 \times 41$ (1,681 bins) with wrapped boundaries (to avoid edge artifacts). We then analytically computed the forward (noising) distributions using Equation 4 (Figure 2A) and the reverse sampling distributions using Equation 10 (Figure 2B) (we did not train any neural networks). As for the noise families, we considered multiple options to test whether the process is indeed robust to noise type. Specifically, we considered Gaussian noise with a linear variance schedule $\sigma_t = 0.03 + 0.04 * (t/T)$ (Figure 2A-B), a bimodal noise kernel consisting of a mixture with two modes symmetrically positioned 0.07 units away from the center resulting in diagonal smearing (Figure 2C), a fade-type noise where there is an increasing probability $p_t = 0.01 + 0.99 * (t/T)$ to sample from a different pre-specified distribution, namely, uniform (Figure 2D) and Gaussian mixture (Figure 2E). $T$ denotes the total number of steps as before (See Supplementary Section B for additional details about the simulations). As can be clearly seen, in all of these cases, whether we changed the noise kernel or the stationary noise distribution, the sampler was able to converge on a good approximation for the underlying Swiss-roll distribution (see Supplementary Figure S1 for additional simulations with the posterior-only sampling scheme and Supplementary Figure S2 for examples of bad samplers due to inadequate noise schedules).

Next, we wanted to test how the approximation accuracy of the sampler depends on the graduality of the noise injection in the forward process as captured by the number of diffusion steps. To that end, we restricted ourselves to the Gaussian case and considered different noise schedules by varying the number of steps interpolating between two fixed levels of variance, that is, $\sigma_t = 0.01 + 0.04 * (t/T)$ for different values of the number of steps $T$ (Figure 3A). To quantify accuracy, we computed the reconstruction error as measured by the KL divergence between the generated data distribution and the true data distribution, i.e., $D_{KL}[q_d(x_0)||p_s(x_0)]$ for each of the chosen schedules (Figure 3B). We also added a measure of inversion "complexity" which measures the biggest change between consecutive distributions along the forward diffusion path, that is, $\max_t D_{KL}[q_{t+1}(x)||q_t(x)]$. The idea is that the bigger this number is the less gradual (more abrupt) the noise injection is, making the learning of the denoiser harder in practice, i.e., recovering a clean sample from noisy data (Figure 3C). As predicted, we see that adding more steps results in lower reconstruction error and lower complexity. In addition, we also see that beyond a certain number of steps the accuracy of the sampler saturates, presumably because the forward process has enough time to reach stationarity and to do that in a smooth way (see also Figure S3 for another similar simulation with a different noise type).

Finally, we also performed diffusion experiments with deep neural networks to validate that this dependence on the number of steps indeed occurs. Specifically, we trained a Denoising Diffusion Probabilistic Model (Ho et al., 2020) to denoise MNIST (LeCun & Cortes, 2005), FMNIST (Xiao et al., 2017), KMNIST (Clanuwat et al., 2018), and CIFAR10 (Krizhevsky et al., 2009) images[1]. For this model, the noise at step $t$ depends on the diffusion parameter $\beta_t = \beta_{min} + (\beta_{max} - \beta_{min}) * t/T$. To investigate the effect of the noise schedule, we set $\beta_{min} = 0.0001, \beta_{max} = 0.02$ and retrained the model multiple times with a different number of total steps each time ($T \in [50, 500]$). To evaluate the sample quality from each trained model, we generated 6,000 images using the same number of steps as the model was trained on, and computed the Fréchet inception distance (FID; Heusel et al. (2017)) between each set of generated images and the training set. Smaller FID values correspond to better image generation quality. We report the results in Figure 4A alongside a sample of the resulting images for different values of $T$ (Figure 4B). We see that similar to the idealized case, gradual noise schedules with a bigger number of steps tend to improve sample quality and reach some saturation level beyond a certain number of steps. We should note that the number of steps needed for high quality samples may be reduced by an appropriate choice of a sampling method, however, the overall dependence on the number of steps remains the same (see e.g., Tables 1-4 and Figures 1, A.1-3 of Watson et al. (2021)).

## 5 DISCUSSION

In this work, we derived a correspondence between diffusion models and an experimental paradigm used in cognitive science known as serial reproduction, whereby Bayesian agents iteratively encode and decode stimuli in a fashion similar to the game of telephone. This was achieved by analyzing the structure of the underlying optimal inversion that the model was attempting to approximate and integrating it in a diffusion sampling process. Crucially, this allowed us to provide a new theoretical understanding for key features of diffusion models that challenged previous formulations, specifically, their robustness to the choice of noise family and the role of noise scheduling. In addition, we validated these theoretical findings by simulating the optimal process as well as by training real diffusion models.

We conclude with some remarks regarding promising future directions. First, in this work we focused on probabilistic diffusion models, as these are the most common. Nevertheless, a new class of models suggests that it is possible to implement completely deterministic diffusion processes (Bansal et al., 2022). A related formulation of serial reproduction (Griffiths & Kalish, 2007) where the assumption of probability matching is replaced with deterministic maximum-a-posteriori (MAP) estimation such that the Gibbs sampling process becomes an Expectation-Maximization algorithm (EM) could provide a suitable setup for reinterpreting such diffusion models. Second, in the present work we assumed asymptotic stationarity (Sohl-Dickstein et al., 2015) as we wanted to focus on the simplest theoretical setup that would allow addressing the question of robustness to noise types.

---

[1] We used Brian Pulfer's PyTorch re-implementation of DDPM – `https://github.com/BrianPulfer/PapersReimplementations`

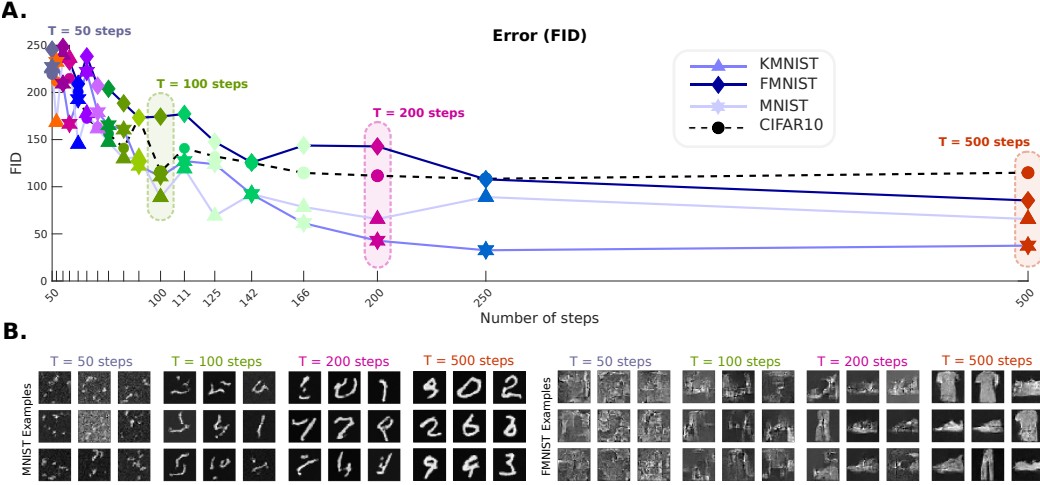

Figure 4: Reconstruction error of a DDPM trained on various datasets for different noise schedules (as captured by the total number of steps interpolating between two noise levels in the forward process). **A.** Performance quantified using FID score. **B.** Example samples from different models.

However, recent work suggests that this assumption too can be relaxed (Watson et al., 2021); a follow-up study could extend our work to include such setups. Third, future work could use this new perspective as an inspiration for defining better noise scheduling metrics by directly trading off intermediate stationarity, convergence rate, and sample accuracy. Finally, this interpretation also suggests that it may be possible to come up with multi-threaded sampling procedures that incorporate multiple serial processes with different learned 'priors' and selection strategies as a way of generating samples that possess a collection of desired qualities. This is inspired by the idea that serial reproduction can be interpreted as a process of cumulative cultural evolution whereby a heterogeneous group of agents jointly reproduce and mutate stimuli so as to optimize them for different properties (Xu et al., 2013; Thompson et al., 2022; van Rijn et al., 2022). We hope to explore these avenues in future research.

**Reproducibility Statement.** The Matlab code for reproducing all simulation analyses of the optimal diffusion sampler discussed in Section 4 is provided with the Supplementary Material attached to this submission. As for necessary code for reproducing the training of the DDPM model, we simply used Brian Pulfer's PyTorch re-implementation of DDPM that can be found at — https://github.com/BrianPulfer/PapersReimplementations. Further details can be found in Appendix B.

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

APPENDIX

## A    DERIVATION OF EQUATION 8

To get to Equation 8, we simply plug Equation 7 into Equation 6 which yields

$$
\begin{aligned}
K &= \sum_{t=1}^{T} \int q(x_0, \dots, x_T) \log \left[ \frac{p(x_{t-1}|x_t)}{q(x_t|x_{t-1})} \right] dx_0 \dots dx_T - H_{q_n} \\
&= \sum_{t=1}^{T} \int q(x_0, \dots, x_T) \log \left[ \frac{p(x_{t-1}|x_t)q(x_{t-1})}{q(x_{t-1}|x_t)q(x_t)} \right] dx_0 \dots dx_T - H_{q_n} \\
&= \sum_{t=1}^{T} \int q(x_0, \dots, x_T) \log \left[ \frac{p(x_{t-1}|x_t)}{q(x_{t-1}|x_t)} \right] dx_0 \dots dx_T + C_q
\end{aligned} \tag{17}
$$

where we defined $C_q = \sum_{t=1}^{T} \int q(x_0, \dots, x_T) \log(q(x_{t-1})/q(x_t)) - H_{q_n}$ which is simply a constant with respect to $p$. Next, observe that the Markovian decomposition in Equation 4 implies that

$$
\int q(x_0, \dots, x_T) dx_0 \dots dx_{t-1} dx_{t+2} \dots dx_T = q(x_t|x_{t-1})q(x_{t-1}) \tag{18}
$$

where $q(x_{t-1}) = \int q(x_{t-1}|x_{t-2}) \dots q(x_1|x_0)q_n(x_0) dx_0 \dots dx_{t-2}$ is the marginal at step $t-1$. By combing this with Bayes formula in Equation 7 we can write

$$
\begin{aligned}
K &= \sum_{t=1}^{T} \int q(x_t|x_{t-1})q(x_{t-1}) \log \left[ \frac{p(x_{t-1}|x_t)}{q(x_{t-1}|x_t)} \right] dx_{t-1} dx_t + C_q \\
&= -\sum_{t=1}^{T} \int q(x_{t-1}|x_t)q(x_t) \log \left[ \frac{q(x_{t-1}|x_t)}{p(x_{t-1}|x_t)} \right] dx_{t-1} dx_t + C_q \\
&= -\mathbb{E}_{x_t \sim q} D_{KL} \left[ q(x_{t-1}|x_t)||p(x_{t-1}|x_t)] \right] + C_q
\end{aligned} \tag{19}
$$

which is the desired result. It's worth noting that a similar result can be found in Ho et al. (2020).

## B    ADDITIONAL SIMULATION DETAILS

### B.1    SIMULATIONS WITH IDEALIZED MODEL

We ran idealized simulations in two dimensions $(x, y)$ in the range $-0.5 \leq x, y \leq 0.5$. We used a 41 by 41 finite grid (1,681 bins) with bin width of 0.025. For the data distribution we used a Swiss-roll distribution similar to Sohl-Dickstein et al. (2015). We made training a bit more challenging by avoiding bins that have zero density. This was done by interpolating the Swiss-roll distribution $p$ with a uniform distribution over the finite grid $p_u$ so in practice we used $p' = 0.9 \cdot p + 0.1 \cdot p_u$. We computed marginal distributions as vectors over the 1,681 bins, and conditional distributions as 1,681 by 1,681 matrices. In all cases, we computed noise terms with wrapped boundaries so that boundary artifacts are avoided. We used three types of noise: (1) Gaussian noise, (2) bimodal noise and (3) fade noise. The Gaussian noise over $(x, y)$ was defined as $p(y, x) = C \cdot \exp\left(-\frac{1}{2}(y-x)^T \Sigma (y-x)\right)$, where $\Sigma = \sigma \cdot I_2$ is a diagonal matrix, $\sigma$ is the noise parameter, and $C$ is a normalization constant. The bimodal distribution was defined as follows: $p(y, x) = C \cdot (\exp\left(-\frac{1}{2}(y-x+\mu)^T \Sigma (y-x+\mu)\right) + \exp\left(-\frac{1}{2}(y-x-\mu)^T \Sigma (y-x-\mu)\right))$, where $\mu = (0.05, 0.05)$ and $\Sigma = \sigma \cdot I_2$. The fade noise depended on a parameter $f$ between 0 and 1, and linearly interpolated between not changing the distribution and completely changing it to some final distribution $p_F$. It was defined by the following formula: $p(y, x) = (1-f) \cdot \delta(y-x) + f \cdot p_F$. Where $\delta(z)$ is the Dirac two-dimensional delta function. For stationary distribution we chose either uniform (Fig. 2D) or a mixture with three modes (Fig. 2E). The three equally weighted modes were located in points $x_1 = (0.1, 0.2)$, $x_2 = (-0.2, -0.1)$, $x_3 = (0.2, -0.2)$ and had covariance matrix of $0.0128 * I_2$, $0.0128 * I_2$, and $0.0128 * I_3$, where $I_2$ is the 2 by 2 identity matrix and $I_3$ is the matrix $[2, 1; 1, 2]$. We provide the code reproducing all figures and simulation as part of the Supplemental materials.

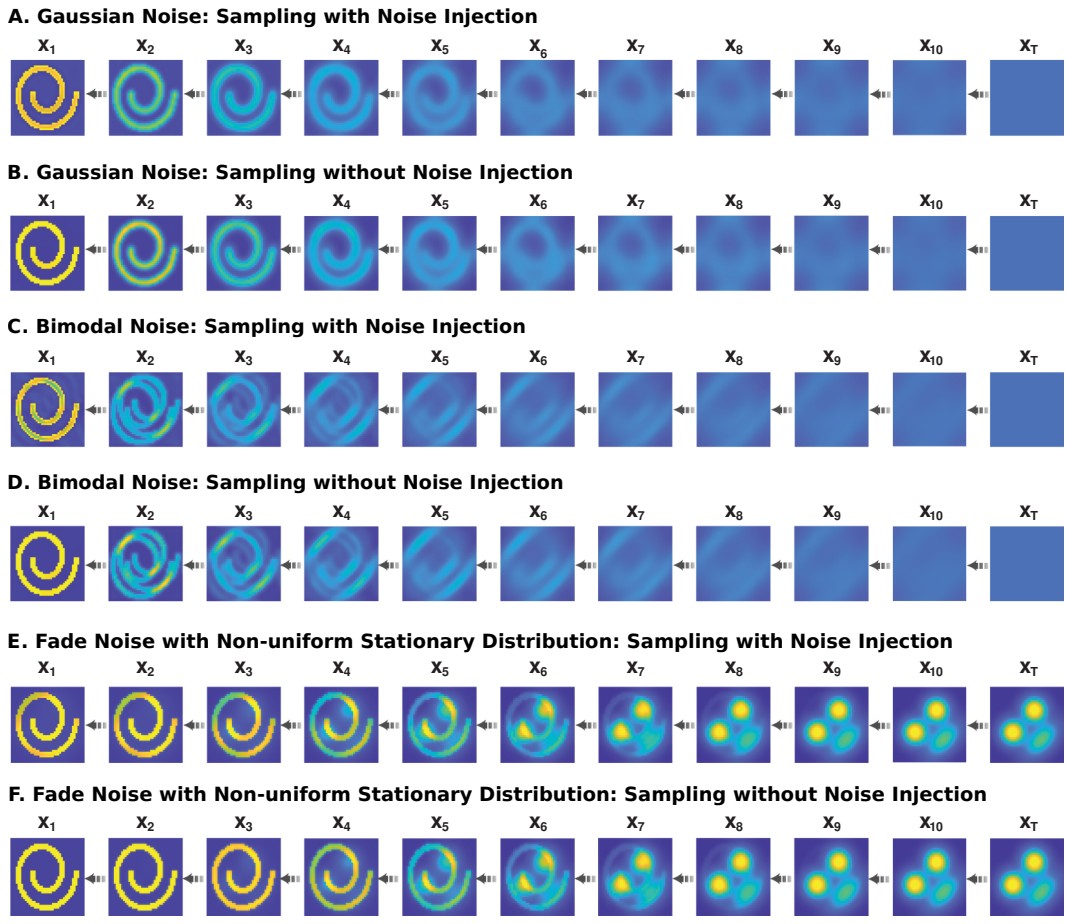

Figure S1: Comparison between sampling with noise injection and without noise injection for different noise types.

## B.2 SIMULATIONS WITH DDPM

For our experiments with deep neural networks, we trained a Denoising Diffusion Probabilistic Model (Ho et al., 2020) to denoise MNIST (LeCun & Cortes, 2005), FMNIST (Xiao et al., 2017), KMNIST (Clanuwat et al., 2018), and CIFAR10 (Krizhevsky et al., 2009) images. We used Brian Pulfer's PyTorch re-implementation of DDPM – (https://github.com/BrianPulfer/PapersReimplementations). The UNet (Ronneberger et al., 2015) used by this implementation is designed to be compatible with single-channel images of size 28x28 (which is standard for MNIST variants) so CIFAR10 images first had to be resized and transformed to grayscale. Our focus is not on the technical implementation of DDPM, so we direct interested readers to Brian Pulfer's repository as it contains helpful documentation and commentary. The key detail is that for this model, the noise at step $t$ depends on the diffusion parameter $\beta_t = \beta_{min} + (\beta_{max} - \beta_{min}) * t/T$ where $T$ is the total number of steps and $t = 0, 1, 2, ..., T$. To investigate the effect of the noise schedule, we set $\beta_{min} = 0.0001, \beta_{max} = 0.02$ and retrained the model multiple times with a different number of total steps each time ($T \in [50, 500]$). For each dataset, this process took less than 2 hours to run on a single RTX 3080 Laptop GPU. To evaluate the sample quality from each trained model, we generated 6,000 images using the same number of steps as the model was trained on, and computed the Fréchet inception distance (FID; Heusel et al. (2017)) between each set of generated images and the training set of the respective dataset.

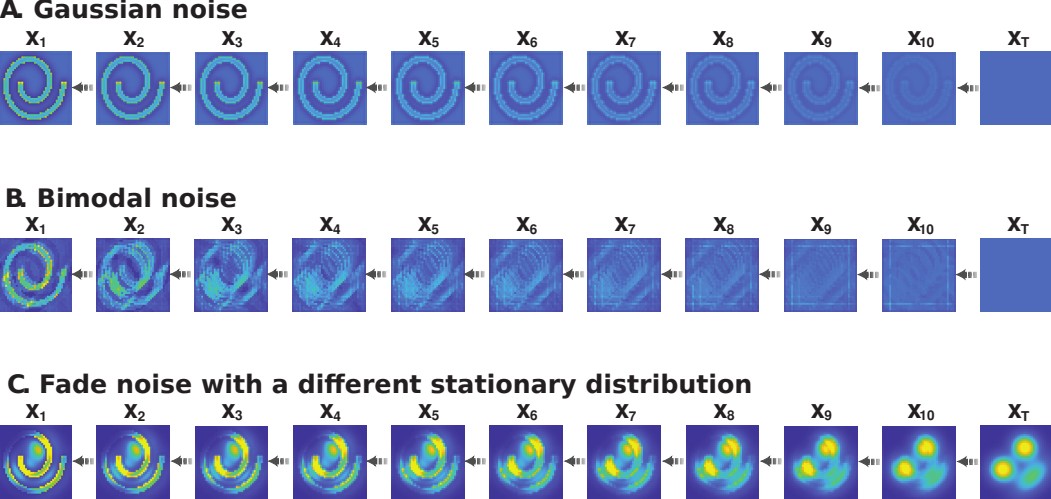

Figure S2: Examples of bad noise schedules. Similar to the other simulations we had $T = 10$ steps, but unlike before the schedules here have a fixed noise parameter that did not change over the 10 steps. **A**. Gaussian noise. We used a constant $\sigma = 0.01$ in all steps. **B**. Bimodal noise. We used a constant $\sigma = 0.001$ in all steps. **C**. Fade noise. We used a constant $f = 0.0001$ in all steps. (see Supplementary Section B, for the definitions of these parameters).

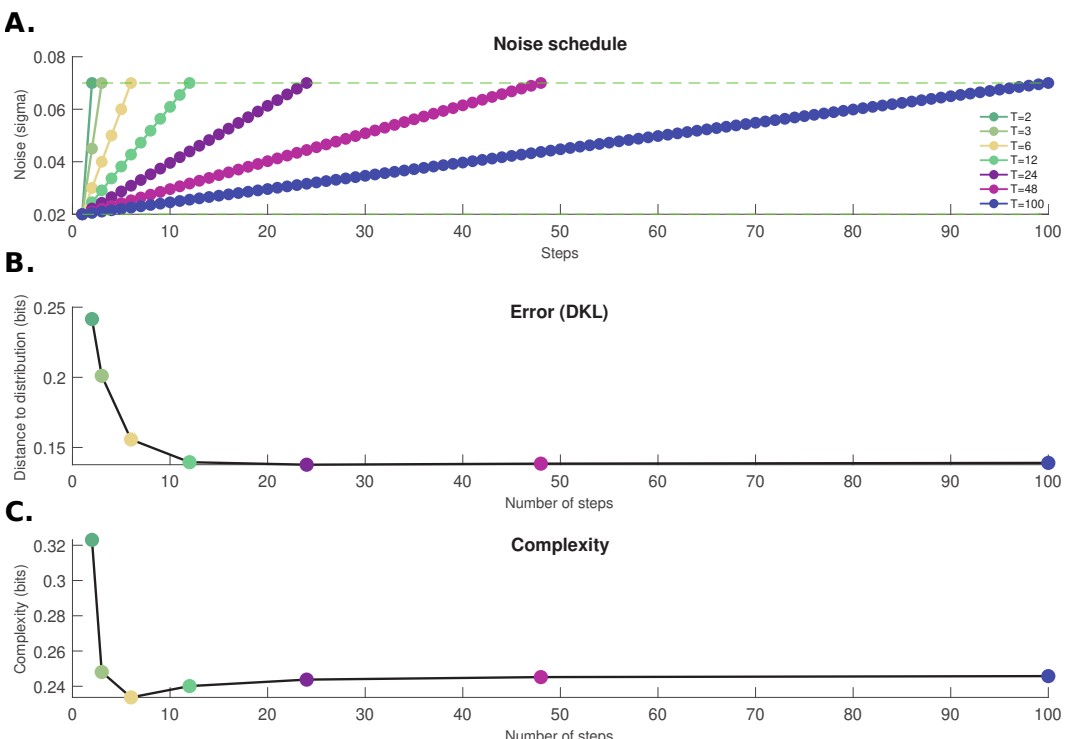

Figure S3: Effect of noise schedule on reconstruction error for an additional bimodal noise type.

