# OpenReview forum: "Analyzing diffusion as serial reproduction"
_ICLR.cc/2023/Conference — Submitted to ICLR 2023_

### Official Review · Reviewer_g85r · 2022-10-23

**Confidence:** 3
**Correctness:** 2
**Technical Novelty And Significance:** 2
**Empirical Novelty And Significance:** 2
**Recommendation:** 5

**Clarity, Quality, Novelty And Reproducibility:**

- Clarity: The notation used in the paper is a little bit difficult to follow. For example, the mathematical form of $T_{q_n}(x_t|x_{t-1},\beta_t)$ is not explained in details and $\beta$ is mentioned as diffusion parameters and strength parameter in different place.
- Quality: I have raised some concerns about technical soundness. Please see my comments above.
- Novelty: The connection discovered by the paper is novel.
- Reproducibility is ok.

**Strength And Weaknesses:**

Strength:

- Theoretical understanding of diffusion probabilistic models is of great research interest recently.
- The connection between the concept of serial reproduction and DPMs is also interesting.

Weakness:
- Contribution: It is not very clear what exactly is the contribution of the paper. The paper introduces a connection between the sampling process of DPMs and serial reproduction. Drawing inspiration from this connection, some key features of diffusion models such as noise scheduling can be principally improved. However, the conclusion that "the noise schedule should be chosen such that it alters the input distribution gradually" is not new (Dhariwal & Nichol, 2021 https://arxiv.org/abs/2105.05233).
- Technical soundness:
    * On page 4, between Eq.(9) and Eq.(10), the paper describes the sampling process of a DPM as $x_t \rightarrow \hat{x}_t  \rightarrow  \cdots$. Why a noisy version of $x_t$, i.e., $\hat{x}_t$ is needed in the sampling process? This seems not true for the well-understood DPMs where the sampling process involves only the optimal posterior distribution (as in Eq.(5)).
    * Latter derivations all depend on the above-mentioned assumption of the sampling process. So it should be crucial to clarify whether the standard sampling process resembles serial reproduction and what Eq.(10) actually means.

Minors:
- In section 2.2, the statement "the noise distribution $q_n$ is stationary" is not always true. For variance exploding diffusion (Song et al. 2020), the noise distribution is not stationary since the variance does not converge.

**Summary Of The Paper:**

This work discusses the connection between the sampling process of diffusion probabilistic models and a cognitive paradigm known as serial reproduction. By building the correspondence, the work tries to provide new theoretical understandings of DPMs and intuitions of improvement of key features of DPMs.

**Summary Of The Review:**

In conclusion, I have raised several concerns regarding the contribution and technical soundness of the proposed method, which prevents me from giving this paper a higher rating now. I will consider increasing my rating if the authors address my concerns and clarify my misunderstandings.

---

> ### Author Response · Authors · 2022-11-18
> **Response to Reviewer g85r**
>
> We thank the reviewer for their careful evaluation of our work as it helped us improve the clarity of our paper, especially in the context of previous literature on DPMs. We will address each of the raised issues in what follows.
>
> ### Reviewer g85r - Addressing concerns regarding technical soundness
> *“On page 4, between Eq.(9) and Eq.(10), the paper describes the sampling process of a DPM as xt→x^t→⋯. Why a noisy version of xt, i.e., x^t is needed in the sampling process? This seems not true for the well-understood DPMs where the sampling process involves only the optimal posterior distribution (as in Eq.(5)).”*
>
> **Answer**: we thank the reviewer for raising this important but subtle issue. We used this formulation of the sampling process to be consistent with recent work that generalizes diffusion processes to broader noise families (e.g., Cold Diffusion, Bansal et al. 2022, https://arXiv:2208.09392; Soft Diffusion, Daras et al. 2022, https://arxiv.org/abs/2209.05442). In these papers, the sampling procedure is decomposed into an alternation between a reconstruction (denoising) operator and a degradation operator (noising) (see Section 3.2 in Bansal et al.). Crucially, this formulation does not contradict the standard Gaussian DDPM formalism and is in fact equivalent to it. We added a new dedicated subsection that establishes this equivalence (Section 3.2) and we reproduce it here for completeness:
>
> “Before moving on to the empirical analysis, we complete our theoretical derivation by showing that our definition of the sampling process is equivalent to the one used in the well-studied Gaussian DDPM formalism of Ho et al. (2020). As noted in that paper (Section 3.2 of Ho et al. (2020)), the posterior is taken to be of the form $p(x_{t-1}|x_t)=\mathcal{N}(x_{t-1};\mu(x_t,t),\Sigma(x_t,t))$ with $\Sigma(x_t,t) = \sigma_t^2 I$, where $\mu(x_t,t)$ is a trainable function and $I$ is the identity matrix. This can be also written as $x_{t-1} = \mu(x_t,t) + \sigma_t z$ where $z\sim\mathcal{N}(0,I)$. In other words, the posterior is given by a Gaussian distribution around a function of the input $\mu(x_t,t)$ with some diagonal covariance matrix with variance $\sigma_t^2$. Intuitively, equivalence then follows from the fact that introducing an additional noising step in the sampling process is simply adding Gaussian noise to a Gaussian posterior which corresponds to a redefinition of the mean and variance parameters (which are design parameters; Dhariwal and Nichol, 2021). More explicitly, our sampling process is defined as x(T) --> x^(T) --> x(T-1) --> ... --> x(0), that is, we start from an initial sample $x_T$ and then successively add noise to it and then denoise it with the posterior. Now, since the initial $x_T\sim q_n(x_T)$ is sampled from the stationary noise distribution, adding noise to it (i.e. transitioning to x^(T)) does not change its distribution so we can equivalently start by denoising $x_T$ using the posterior (as in DDPM) and then adding noise and so on. Mathematically, this corresponds to applying to the generated posterior sample (i.e., the denoised $x_T$) a generic noisy transformation of the form $x \rightarrow \alpha x + \sigma z'$ where $z'\sim \mathcal{N}(0, I)$ where $\alpha$ and $\sigma$ are some scaling and variance parameters (similar to Equation 2 in Ho et al., 2020). Now, when combined with the Gaussian posterior above this yields $x_{T-1} = \alpha\mu(x_T,T) + \alpha \sigma_T z + \sigma z'$ which corresponds to $p(x_{T-1}|x_{T}) = \mathcal{N}(x_{T-1};\alpha\mu(x_T,T), (\alpha^2\sigma_T^2 + \sigma^2) I )$, but this is equivalent to the formula used in  Ho et al. (2020) up to a redefinition of the mean and variance, namely, $\mu_T(x_T,T)\rightarrow \alpha\mu(x_T,T)$ and $\sigma_T^2 \rightarrow \alpha^2\sigma_T^2 + \sigma^2$. The same holds for all subsequent steps which, as in the case of DDPM, terminate with a final application of the posterior denoiser. Thus, we see that the two definitions are equivalent up to a redefinition of the trainable posterior parameters.”
>
> For completeness, we updated our simulations to also include the posterior-only sampling scheme (i.e., skipping the noising step, but keeping the other aspects of the simulation). The results were very similar to the ones from the scheme used in the paper (see Figure S1).
>
> **(see part 2 of response below)**

---

> > ### Author Response · Authors · 2022-11-18
> > **Response to Reviewer g85r (part 2)**
> >
> > ### Reviewer g85r - Clarifying the contributions
> > *“Drawing inspiration from this connection, some key features of diffusion models such as noise scheduling can be principally improved. However, the conclusion that "the noise schedule should be chosen such that it alters the input distribution gradually" is not new (Dhariwal & Nichol, 2021 https://arxiv.org/abs/2105.05233).”*
> >
> > **Answer**: We already mentioned another paper that make a similar statement (Daras et al., 2022), but we agree that Dhariwal & Nichol, 2021 is relevant, we now revised the section about this issue and cite both papers. While it is true that the importance of a smooth noise schedule for sampling quality has been observed in these papers, we believe that the main contribution of our work is a framework that can *jointly* explain this finding and the robustness to different noise families. To our knowledge, we are the first to provide such a principled explanation. As pointed out in our paper and further iterated by another reviewer, this is essential due to the fact that existing theoretical treatments of DPMs assume Gaussian noise which renders them incomplete in light of the aforementioned findings. We now clarify this in the introduction:
> >
> > “The main contribution of the present paper is to make the correspondence between diffusion and serial reproduction precise and show how the observed properties of diffusion models, namely, their robustness to the choice of noise family, and the role of adequate noise level scheduling for good sampling, jointly arise as a natural consequence.”
> >
> > More broadly, our work connects diffusion models that are currently the topic of active research to a very well studied cognitive process with a large literature. We believe that exploring other implications of this correspondence could lead to more interesting insights and applications as we describe at the end of the Discussion section which we reproduce below for completeness:
> >
> > “Finally, this interpretation also suggests that it may be possible to come up with multi-threaded sampling procedures that incorporate multiple serial processes with different learned ‘priors’ and selection strategies as a way of generating samples that possess a collection of desired qualities. This is inspired by the idea that serial reproduction can be interpreted as a process of cumulative cultural evolution whereby a heterogeneous group of agents jointly reproduce and mutate stimuli so as to optimize them for different properties (Xu et al., 2013; Thompson et al., 2022; van Rijn et al., 2022).”
> >
> > ### Reviewer g85r - Addressing other minor comments
> > *“In section 2.2, the statement "the noise distribution qn is stationary" is not always true. For variance exploding diffusion (Song et al. 2020), the noise distribution is not stationary since the variance does not converge.”*
> >
> > **Answer**: We thank the reviewer for bringing this paper to our attention. In our paper, we chose to work under the assumptions proposed in the original diffusion paper by Sohl-Dickstein et al. (2015) and focus on the robustness to noise. To maintain transparency, however, we now acknowledge in the text that there are variants for which this assumption does not hold.
> >
> > *“Clarity: The notation used in the paper is a little bit difficult to follow. For example, the mathematical form of Tqn(xt|xt−1,βt) is not explained in details and β is mentioned as diffusion parameters and strength parameter in different place.”*
> >
> > **Answer**: we revised the manuscript and improved the clarity of notation.

---

> > > ### Author Response · Authors · 2022-12-03
> > > **Response to Reviewer g85r - follow up**
> > >
> > > We just wanted to follow up and make sure the reviewer saw our response. In particular, we want to draw their attention to our response to their concern about the technical soundness of our definition of the sampling process. This was an important point and we clarified it by adding a new section to the paper (also provided in our response above). We hope to hear back from the reviewer soon!

---

> > ### Comment · Reviewer_g85r · 2022-12-08
> > **One more question**
> >
> > Thank you for the detailed responses!
> >
> > I have one further question about the interpretation of the sampling process. I agree that the additional noising step can be interpreted as a part of the original backward distribution $p(x_{t-1}|x_t)$. However, if the variance of the Gaussian backward distribution is split into two parts, where one of them corresponds to the noise in x^(T), then $p_{t-1}$ in Eqn. (10) no longer corresponds to the origin $p_{t-1}(x_{t-1}|x_t)$ in the diffusion model. Does it have any influence on the theory and results of the paper? Or it doesn't really matter how much noise we give to the additional noising step?
> >
> > The author has clarified the contribution of the paper and thus I raise my score to 5. Pointing out the correspondence between diffusion and serial reproduction is interesting. However, the conclusion we can draw from this correspondence, such as the robustness of the choice of noise family, and the fact of adequate noise level scheduling resulting in good sampling are not new to the community.

---

> > > ### Author Response · Authors · 2022-12-09
> > > **Response to additional question**
> > >
> > > Thank you for raising our score and for the thoughtful comments.
> > >
> > > Again, you asked a great question. We added a short note in the section “Clarifying the connection to previous work ” that describes this issue briefly, and we also added a more detailed simulation of this point in the Appendix.  We agree that this is an interesting and valid point, and that answering it adds to the theoretical implications of the paper.
> > >
> > > You are right, if indeed the noise injection is chosen in an arbitrary way, then there is potentially a mismatch between the injected noise magnitude and the associated denoising step in the sampling process. Our theory suggests that this is clearly not desirable, as the analytically elegant solution whereby the sampling process yields the true data distribution relies on the detailed balance condition (Eq. 11) which is only true when the noise magnitudes match between denoising and noising steps. Nevertheless, the case of small mismatch between noise magnitudes was discussed to a certain extent in the cognitive literature and has been shown not to alter the resulting sampling distribution significantly (see Griffiths and Kalish, 2007; Xu and Griffiths 2010; Jacoby and McDermott, 2017). This is also consistent with the observation made by the authors of the DDPM paper (Section 3.2 of Ho et al. 2020) that trying two related schemes for choosing the variances of the denoiser resulted in similar performance.
> > > To further test this hypothesis, we simulate the effect of having this mismatch between the noise injection magnitudes and the denoising. The resulting stationary distribution is still quite similar but the results are slightly deteriorated (in terms of the KL divergence). We now added an additional simulation to the Appendix (and refer to it in the section “Clarifying the connection to previous work”).
> > >
> > > **References:**
> > >
> > > Xu, Jing, and Thomas L. Griffiths. “A rational analysis of the effects of memory biases on serial reproduction.” Cognitive psychology 60, no. 2 (2010): 107-126.
> > >
> > > Griffiths, Thomas L., and Michael L. Kalish. "Language evolution by iterated learning with Bayesian agents." Cognitive science 31, no. 3 (2007): 441-480.
> > >
> > > Jacoby, Nori, and Josh H. McDermott. "Integer ratio priors on musical rhythm revealed cross-culturally by iterated reproduction." Current Biology 27, no. 3 (2017): 359-370.

---

### Official Review · Reviewer_fE3L · 2022-10-24

**Confidence:** 4
**Correctness:** 3
**Technical Novelty And Significance:** 2
**Empirical Novelty And Significance:** 2
**Recommendation:** 6

**Clarity, Quality, Novelty And Reproducibility:**

This paper provides a new way of understanding why diffusion models work so well. There are several equations can be modified to align with existing DDPM mathematical equations.
The clarity and quality are average.
The novelty is limited and can be further improved.
The reproducibility of this paper is high since it is based on existing open-source code.

Detailed questions and comments:

1. Equation (5), better attach $\theta$ for $p$ to indicate that it is trainable with parameter set and differs with $q$ in equation (4).

2. Equation (8), $q$ better with a given sample $x_0$, e.g., from $q(x_{t-1}|x_t)$ to $q(x_{t-1}|x_t, x_0)$, to express the posterior distribution in the forward diffusion process.

3. Do you have more results of simulation on larger datasets besides Mnist dataset? Figure 4 shows some results – they are largely influenced by the sampling methods you use as well, such as Euler-Maruyama, the predictor-corrector and ODE sampling. Currently, the T=50 vs. T=500 is significantly different and based on my own experiments, T=50 could yield quite good results with better sampling methods and without changing anything of the trained model.


**Strength And Weaknesses:**

Strong:
1. The analysis and connection of between diffusion models and serial reproduction paradigm;

2. Simulations on datasets to show the robustness of applying different types of noise distributions.


Weak:
1. Larger scale datasets and richer experiments can help improve the quality of this paper;

2. The connection between diffusion models and serial reproduction paradigm seems to have less novelty in terms of methodology and it is difficult to say that new knowledge is obtained after reading this article.


**Summary Of The Paper:**

This paper builds connections between existing diffusion models such as DDPM with serial reproduction paradigm. One major conclusion is that, current diffusion models can be explained as a natural consequence of this connection correspondence. Simulations on the MNIST dataset shows that the connection can be utilized for richer noise distribution family such as bimodal and fade.

**Summary Of The Review:**

Given current strengthes and weaknesses, it is difficult to score a high recommendation to this paper. Some equations can be updated, richer datasets are preferred and more sampling algorithms can be used to perform rich ablation study.

--

I would like to rank this paper higher, after reading their detailed responses.
Most of my questions were well answered.
Again, if any larger scale datasets' results are reported or shared, this paper worth an even higher score.

---

> ### Author Response · Authors · 2022-11-18
> **Response to Reviewer fE3L**
>
> We are grateful for the reviewer’s detailed and constructive comments and have made efforts to address them. We detail these efforts below.
>
> ### Reviewer fE3L - Addressing comments on derivation
> *“There are several equations can be modified to align with existing DDPM mathematical equations.”*
>
> **Answer**: We have revised the manuscript to make the alignment with the existing DDPM mathematical formulation clearer. In particular, we added a dedicated section (Section 3.2) that establishes the equivalence with the DDPM formalism explicitly. We also addressed the individual comments on the equations below.
>
> *“Equation (5), better attach θ for p to indicate that it is trainable with parameter set and differs with q in equation (4).”*
>
> **Answer**: We have updated the manuscript accordingly.
>
> *“Equation (8), q better with a given sample x0, e.g., from q(xt−1|xt) to q(xt−1|xt,x0), to express the posterior distribution in the forward diffusion process.”*
>
> **Answer**: These two formulations are equivalent. We show in the Appendix (Eqs. 17-19) that the dependence on $x_0$ can be eliminated. While this form is not tractable in practice, it is useful for the analysis that follows. A very similar argument can be found in Ho et al. (2020) DDPM paper where they show (Appendix A, Eqs. 17-26 in their paper) that the familiar variational bound with $x_0$ (Eq. 5 in their paper) can be recast into a form without it (Eq. 16 in their paper).  To further clarify this we added a clarification to the paper:
>
> “While Equation (8) is not necessarily the most tractable form of the bound $K$, it will prove useful in the next section when we make the connection to serial reproduction (see Appendix A, Eqs. 17-26 in Ho et al. (2020) for a similar derivation).”
>
>
> ### Reviewer fE3L - Addressing comments regarding other datasets
> *“Do you have more results of simulation on larger datasets besides Mnist dataset?*
>
> **Answer**: We have now included results for three additional datasets, namely, CIFAR10, FMNIST, and KMNIST all exhibiting similar trends. More explicitly, we updated Figure 4 to include the error (FID) as a function of steps for the additional datasets. We see that, as in the case for MNIST, the error gradually decreases and saturates as a function of the steps for all four datasets.
>
> *“Figure 4 shows some results – they are largely influenced by the sampling methods you use as well, such as Euler-Maruyama, the predictor-corrector and ODE sampling. Currently, the T=50 vs. T=500 is significantly different and based on my own experiments, T=50 could yield quite good results with better sampling methods and without changing anything of the trained model.”*
>
> **Answer**: While we agree that the choice of sampler can greatly improve the quality of the results for smaller values of T, our argument is that the overall quality of sampling should improve as the number of training/sampling steps increases, independent of choice of sampler. This can be seen, for example in Tables 1-4 and Figures 1, A.1-3 of Watson et al. (2022; https://arxiv.org/pdf/2202.05830.pdf) where the authors explored a large variety of samplers and their performance at different values of T (denoted by K in the tables). As the reviewer noted, the performance of different samplers varies compared to each other, but what remains consistent is that every sampler performs better when used with a larger number of steps.  We have updated the manuscript to clarify this point.

---

> > ### Comment · Reviewer_fE3L · 2022-12-08
> > **thank you for the detailed responses**
> >
> > Thank you for the detailed responses and I think most of my questions are well answered and thus I updated my score.
> >
> > Again, only if you can share more results on larger scale datasets, this paper deserves an even higher score.
> > The ideas described are novel and just prefer some large-scale comparisons to learn more evidences of the performance of your strong ideas.

---

> > > ### Author Response · Authors · 2022-12-09
> > > **Response to Reviewer fE3L - follow up**
> > >
> > > Thank you for reading our response and updating your score! As you suggested, we are currently running large-scale experiments with CelebA and ImageNet but they are taking a long time to run. Similar experiments were conducted by Watson et al. (2022; https://arxiv.org/pdf/2202.05830.pdf) and we expect our results will reproduce theirs. We will include the results in the final paper, and post them here as they become available. We hope you might consider updating your score.

---

> > > > ### Author Response · Authors · 2022-12-11
> > > > **Response to Reviewer fE3L - CelebA results**
> > > >
> > > > We are excited to share that our large-scale CelebA experiment results are now ready! We conducted them consistently with our MNIST, KMNIST, FMNIST, and CIFAR10 experiments. Since we cannot share figures here, we summarize the CelebA results in a table below. We find the same overall trend with CelebA as we did for the other datasets: as the number of steps decreases, the quality also decreases (i.e. FID increases).
> > > >
> > > >
> > > > | num_steps | FID                |
> > > > |-----------|--------------------|
> > > > | 1000      | 173.2581453291738  |
> > > > | 500       | 220.73479346227276 |
> > > > | 333       | 142.32501348902662 |
> > > > | 250       | 148.68236464998733 |
> > > > | 200       | 139.4046492392768  |
> > > > | 166       | 172.43193679419122 |
> > > > | 142       | 169.7324752410991  |
> > > > | 125       | 185.76644366040023 |
> > > > | 111       | 183.09939693464753 |
> > > > | 100       | 187.61114814257888 |
> > > > | 90        | 212.59871334075697 |
> > > > | 83        | 216.5169540154675  |
> > > > | 76        | 222.98385762609826 |
> > > > | 71        | 247.92078578988344 |
> > > > | 66        | 248.45585245158082 |
> > > > | 62        | 249.2295063931025  |
> > > > | 58        | 255.8826613806992  |
> > > > | 55        | 256.9601080190655  |
> > > > | 52        | 280.18742133115927 |
> > > > | 50        | 294.7392674497123  |

---

### Official Review · Reviewer_Q4TQ · 2022-10-26

**Confidence:** 4
**Correctness:** 4
**Technical Novelty And Significance:** 4
**Empirical Novelty And Significance:** 2
**Recommendation:** 8

**Clarity, Quality, Novelty And Reproducibility:**

**Clarity**
- My only negative comment on clarity is regarding the use of the term "sampling". I think it is common to distinguish between "learning/training" and "sampling", the latter referring to the denoising process. However, in the introduction, it is said that "sampling is then done using a sequential procedure whereby an input signal is iteratively *corrupted by noise and then denoised*". I found including noising in the sampling unnaturally trying to resemble the explanation on serial reproduction.
- Apart of the previous comment, the paper is well organised, generally well written and easy to read.

**Quality.**
Following up with my previous comment on "sampling", the same idea of adding noise during sampling is used formally in Sec. 3, when describing sampling process as a Markov chain, with terms like, $x_t \rightarrow  \hat{x}_t$. What are these terms? Are they just artefacts for the explanation, meaning that $x_t$ is drawn from some noise kernel? Or it is assumed that $x_t$ is actually corrupted with noise before every denoising step? Please be explicit in the text if this is an artefact to avoid confusion.

**Novelty.**
The paper establishes a novel and insightful connection.

**Reproducibility.**
The paper includes a final section on reproducibility pointing to the code in the supplementary material. While I agree that making the code available is the gold standard for reproducibility, I encourage the authors to describe all the missing details also in the appendix, especially regarding the deep learning experiments for completeness.

**Strength And Weaknesses:**

**Strength**
- The paper establishes a novel and insightful connection with cognitive neuroscience, making it accessible and hopefully inspire other researchers.
- The insights derived from this connection are relevant, as they explain two empirically observed properties, especially the robustness against the noise family, which made previous formulations that assumed Gaussian noise incomplete.

**Weaknesses**
- See comments on clarity below.

**Summary Of The Paper:**

Inspired by a cognitive science paradigm, known as "serial reproduction", the authors show how the diffusion model approximates a Bayesian posterior at every denoising step, with the diffusion kernel as likelihood and the forward marginal as prior. This re-interpretation is enough to explain two properties of diffusion models that have been empirically observed: i) a diffusion model can approximate
the true data distribution irrespective of the choice of noise family for a fine enough noise schedule; and ii) the noise parameter has to diminish in order to reduce reconstruction error. The paper supports the theoretical insights with illustrative experiments, first on a setup where the noising and denoising distributions can be computed analytically and different noise schemes; and second, with MNIST images. One final insight provided by these experiments is that the sample accuracy saturates after some number of steps, and based on the theoretical insights, the authors suggest this is due to the forward process having converged to its stationary distribution.

**Summary Of The Review:**

#### Meta-review
The authors have addressed my concern on the definition of the denoising process, explaining that it follows the latest research on more effective sampling for the more general cold-diffusion processes, and showing an equivalence with the original DDPM derivation.

---
#### First review
The paper provides a novel and insightful formulation of diffusion models, shedding light on two empirical observations that had not clear explanation with previous formulations. This is an interesting contribution that improves our understanding of these powerful models.
My current recommendation is based on the understanding that my comment on quality is just a matter of clarity, and not a misunderstanding or variation of the denoising process.

---

> ### Comment · Reviewer_Q4TQ · 2022-11-17
> **Concern about adding noise during sampling**
>
> I have asked the authors why they included noising steps in the sampling process in my comments on Clarity and Quality. At first, I thought it must be an artefact of their explanation that I might have missed. However, if they do not respond satisfactorily to this critical point, I will assume this is an error in their framework and will reduce my recommendation score.

---

> ### Author Response · Authors · 2022-11-18
> **Response to Reviewer Q4TQ**
>
> We are very grateful for the reviewer’s favorable evaluation of our work. We are also glad that they found the connection to serial reproduction insightful and relevant, and would like to thank them for their valuable comments.
>
> ### Reviewer Q4TQ - addressing concerns regarding the definition of sampling
> *“Following up with my previous comment on "sampling", the same idea of adding noise during sampling is used formally in Sec. 3, when describing sampling process as a Markov chain, with terms like, xt→x^t. What are these terms? Are they just artefacts for the explanation, meaning that xt is drawn from some noise kernel? Or it is assumed that xt is actually corrupted with noise before every denoising step? Please be explicit in the text if this is an artefact to avoid confusion.”*
>
> **Answer**: The reason we chose to define the sampling process in a way that incorporates the addition of noise between applications of the denoiser was to make contact with the recent literature on generalized diffusion models that go beyond Gaussian noise (Cold Diffusion, Bansal et al. 2022, https://arXiv:2208.09392; Soft Diffusion, Daras et al. 2022, https://arxiv.org/abs/2209.05442). These works highlight the idea that sampling (i.e., generation of stimuli) can be thought of as an alternating process between corruption and reconstruction operators (see Section 3.2 in Bansal et al.). Importantly, our definition as provided in the paragraph above Equation (10) is in fact equivalent to the one used in traditional Gaussian DDPMs (Ho et al., 2020) up to a redefinition of the parameters. We now clarify this in the text in two places based on the reviewer’s comments, namely,
>
> –In the introduction:
>
> “Initially, the noise family was restricted to the Gaussian class (Sohl-Dickstein et al., 2015; Song & Ermon, 2019; Ho et al., 2020) and the process was understood as a form of Langevin dynamics (Song & Ermon, 2019). However, recent work showed that this assumption can be relaxed substantially (Bansal et al. 2022; Daras et al. 2022) by training diffusion models with a wide array of degradation families. One feature of this work is that it highlights the idea that sampling (i.e. synthesis) can be thought of more generally as an alternating process between degradation and restoration operators (Bansal et al. 2022). This in turn calls into question the theoretical understanding of these models and necessitates new approaches.”
>
> **(see part 2 of response below)**

---

> > ### Author Response · Authors · 2022-11-18
> > **Response to Reviewer Q4TQ (part 2)**
> >
> > –In a new dedicated section (Section 3.2) that establishes the equivalence:
> >
> > “Before moving on to the empirical analysis, we complete our theoretical derivation by showing that our definition of the sampling process is equivalent to the one used in the well-studied Gaussian DDPM formalism of Ho et al. (2020). As noted in that paper (Section 3.2 of Ho et al. (2020)), the posterior is taken to be of the form $p(x_{t-1}|x_t)=\mathcal{N}(x_{t-1};\mu(x_t,t),\Sigma(x_t,t))$ with $\Sigma(x_t,t) = \sigma_t^2 I$, where $\mu(x_t,t)$ is a trainable function and $I$ is the identity matrix. This can be also written as $x_{t-1} = \mu(x_t,t) + \sigma_t z$ where $z\sim\mathcal{N}(0,I)$. In other words, the posterior is given by a Gaussian distribution around a function of the input $\mu(x_t,t)$ with some diagonal covariance matrix with variance $\sigma_t^2$. Intuitively, equivalence then follows from the fact that introducing an additional noising step in the sampling process is simply adding Gaussian noise to a Gaussian posterior which corresponds to a redefinition of the mean and variance parameters (which are design parameters; Dhariwal and Nichol, 2021). More explicitly, our sampling process is defined as x(T) --> x^(T) --> x(T-1) --> ... --> x(0), that is, we start from an initial sample $x_T$ and then successively add noise to it and then denoise it with the posterior. Now, since the initial $x_T\sim q_n(x_T)$ is sampled from the stationary noise distribution, adding noise to it (i.e. transitioning to x^(T)) does not change its distribution so we can equivalently start by denoising $x_T$ using the posterior (as in DDPM) and then adding noise and so on. Mathematically, this corresponds to applying to the generated posterior sample (i.e., the denoised $x_T$) a generic noisy transformation of the form $x \rightarrow \alpha x + \sigma z'$ where $z'\sim \mathcal{N}(0, I)$ where $\alpha$ and $\sigma$ are some scaling and variance parameters (similar to Equation 2 in Ho et al., 2020). Now, when combined with the Gaussian posterior above this yields $x_{T-1} = \alpha\mu(x_T,T) + \alpha \sigma_T z + \sigma z'$ which corresponds to $p(x_{T-1}|x_{T}) = \mathcal{N}(x_{T-1};\alpha\mu(x_T,T), (\alpha^2\sigma_T^2 + \sigma^2) I )$, but this is equivalent to the formula used in  Ho et al. (2020) up to a redefinition of the mean and variance, namely, $\mu_T(x_T,T)\rightarrow \alpha\mu(x_T,T)$ and $\sigma_T^2 \rightarrow \alpha^2\sigma_T^2 + \sigma^2$. The same holds for all subsequent steps which, as in the case of DDPM, terminate with a final application of the posterior denoiser. Thus, we see that the two definitions are equivalent up to a redefinition of the trainable posterior parameters.”
> >
> > For completeness, we revised our simulations to also include a version with posterior-only sampling scheme (i.e., skipping the noising step, but keeping all other aspects of the simulation the same). The results were very similar to the ones arising from the scheme used in the paper (see Supplementary Figure S1).
> > ### Reviewer Q4TQ - Addressing comments regarding reproducibility
> > *“While I agree that making the code available is the gold standard for reproducibility, I encourage the authors to describe all the missing details also in the appendix, especially regarding the deep learning experiments for completeness.”*
> >
> > **Answer**: we have expanded the Appendix to include all details regarding the experiments.

---

> > > ### Comment · Reviewer_Q4TQ · 2022-11-20
> > > **The authors response has allayed my concerns**
> > >
> > > I thank the authors for the explanation on their iterative noise-denoise steps during the sampling process. I will keep my recommendation score and increase my correctness score accordingly.

---

### Official Review · Reviewer_xQWb · 2022-11-04

**Confidence:** 3
**Correctness:** 2
**Technical Novelty And Significance:** 3
**Empirical Novelty And Significance:** 3
**Recommendation:** 5

**Clarity, Quality, Novelty And Reproducibility:**

The paper is clearly written. The connection to serial reproduction is novel. The mathematical analysis seems new, but it is not precise.

**Strength And Weaknesses:**

I think this is an interesting paper. The main take-away (which is fairly intuitive) is that the corruption should be smooth -- and that's pretty much all that matters to sample from the correct distribution at the end. The main issue I have is that the mathematical statements yielding this result are not very precise, i.e. they do not consider how error propagates in Eq. 15 and the result only holds when the learning is perfect.

Strengths:
* the paper is well-written.
* The simulation experiments are really interesting. They show that for toy-distributions (that we can characterize their density all the way), the corruption type is not really important to reconstruct the proper distribution. I am wondering if that's true for *any* corruption with smooth transitions.
* The topic of the paper is relevant since there have been many recent attempts to generalize diffusion models.


Weaknesses:
* the mathematical statements in the paper are not precise. While I appreciate the fact that the authors try to build intuition, it would be cool to have an analysis of how the errors propagate in Equation 15. This "approximately stationary" statement might completely blow-up the distribution we are sampling from once you integrate over all diffusion steps. How "approximately" is needed to get a reasonable bound?
* the analysis seems to be held only in the case where the solution gives 0 KL. How do learning errors propagate when that's not the case?
* There is no characterization of the conditions under which "approximately stationary" is not too far off. How is this connected to the smoothness of corruption? Are there toy models that satisfy this property? Are there toy models that don't satisfy this property and for them the learning fails?
* the connection to serial reproduction is interesting, yet kind of indirect. For me, the main point is that independent of the corruption type, you still sample from the correct final distribution as long as the condition of Eq. 15 is satisfied. It is true that in serial reproduction you also sample from the priors, independent of the transition kernel and I do appreciate the connection. However, I think it is an overstatement to say that this connection yields many insights on how diffusion models work.


**Summary Of The Paper:**

The paper establishes a connection between diffusion models and serial reproduction. The main result is that as long as the distributions of images along the diffusion time remain approximately stationary, then the perfect solution for minimizing the KL between the sampling process and the real process leads to samples from the correct (clean) distribution. This holds independent of the corruption type, which explains to a certain degree the success of recent diffusion models with more general corruptions.

**Summary Of The Review:**

I think this is a paper in an interesting direction. However, I am concerned because of the lack of clarity in the mathematical statements.

---

> ### Author Response · Authors · 2022-11-18
> **Response to Reviewer xQWB**
>
> We would like to thank the reviewer for their careful evaluation of our work. We found the feedback very useful for improving the quality of our work and we have updated the manuscript accordingly. The details can be found below.
>
> ### Reviewer xQWB - responding to raised points
>
> *“the mathematical statements in the paper are not precise. While I appreciate the fact that the authors try to build intuition, it would be cool to have an analysis of how the errors propagate in Equation 15. This "approximately stationary" statement might completely blow-up the distribution we are sampling from once you integrate over all diffusion steps. How "approximately" is needed to get a reasonable bound?”*
>
> **Answer**: This is indeed a useful analysis that we considered. The fact that the true data distribution $q_d(x)$ pops out of the integral on the right-hand-side of the formula for the sampling distribution $p_s(x)$, i.e., Eq. (12)
>
> $p_s(x_0)=q_d(x_0)\int p_{s,0}(x_1|x_0)\frac{q_1(x_1)}{q_d(x_1)}\dots p_{s,T-1}(x_T|x_{T-1})\frac{q_n(x_T)}{q_{T-1}(x_T)}dx_{1\dots T}$
>
> makes it very tempting to take the KL divergence between $q_d(x)$ and $p_s(x)$ and then plug in Eq. (12) and use some manipulations to link it to the KL divergence between the two sides of Eq. (15) (a direct measure of the deviation from stationarity), which would lead to a bound. Unfortunately, we were not able to derive a simple analytical formula, but we did explore this question empirically in Figure 3. As panels B and C suggest, the error (i.e., $D_{KL}[q_d(x)||q_s(x)]$) appears to closely follow a term that we referred to as “inversion complexity” which is given by the largest KL divergence between two consecutive marginals along the forward pass $max_t D_{KL}[q_{t+1}(x)||q_t(x)]$ (basically the largest violation of stationarity as quantified by the KL between the two sides of Eq. 15, for all steps).
>
> *“the analysis seems to be held only in the case where the solution gives 0 KL. How do learning errors propagate when that's not the case?”*
>
> **Answer**: to answer this question we conducted experiments with real diffusion models that naturally deviate from the optimal 0 KL solution. Figure 4 shows how errors (FID) propagate as a function of the number of steps in the forward process (i.e., the graduality of the noise schedule) for 4 different datasets (originally MNIST, but now also for KMNIST, FMNIST and CIFAR10 upon request from another reviewer). We see that smoother schedules (i.e. those with higher T) result in lower FID values consistent with Figure 3 for the ideal model.
>
> *“There is no characterization of the conditions under which "approximately stationary" is not too far off. How is this connected to the smoothness of corruption? Are there toy models that satisfy this property? Are there toy models that don't satisfy this property and for them the learning fails?”*
>
> **Answer**: Our simulation experiments provide an empirical characterization of approximate stationarity. Figure 2 provides multiple toy models for which this property holds for different noise types and Figure 3 shows how different noise schedules can affect the error between the true distribution and the sampling distribution $D_{KL}[q_d(x)||q_s(x)]$. We also link the error to a property of the forward process that we referred to as “inversion complexity” $max_t D_{KL}[q_{t+1}(x)||q_t(x)]$ (see above). We have added another example of this analysis with a different noise type (Figure S3). In addition, we now also include in Supplementary Figure S2 toy examples for which sampling doesn’t work. In these cases, the noise magnitude in the forward process is not sufficient which in turn results in large sampling errors.

---

> > ### Author Response · Authors · 2022-12-09
> > **Response to Reviewer xQWB - follow up**
> >
> > We just want to make sure the reviewer read our response. Specifically, regarding the addition of examples showing toy models for which sampling does not work (see new Supplementary Figure S2). We hope to hear back from the reviewer soon.

---

### Decision · Program_Chairs · 2023-01-20

**Decision:**

Reject

**Justification For Why Not Higher Score:**

See weaknesses above.

**Justification For Why Not Lower Score:**

N/A

**Metareview: Summary, Strengths And Weaknesses:**

Summary: By observing that diffusion models bear a resemblance to a concept of "serial reproduction" in cognitive science, the paper draws insights into why diffusion models turn out to be robust to the choice of noise family, and what role noise level scheduling plays for effective sampling.

Strengths: Understanding why diffusion models are so effective, and how they are influenced by noise families and noise scales, is a question of significant current interest.  The sim experiments are interesting as they confirm that precise nature of corruption is not really that important.

Weaknesses: The deeper contributions and novelty of this paper remained unclear at the end of the review process. The conclusion that "the noise schedule should be chosen such that it alters the input distribution gradually" is not new. That the connection to serial connection yields many new insights was not entirely convincing in terms of changing the core algorithmic backbones of diffusion models. The reviewers make several suggestions to improve technical presentation and experimental analysis, e.g. some mathematical statements could be made more precise.